# Safety and Efficacy of IL-12 Plasmid DNA Transfection into Pig Skin: Supportive Data for Human Clinical Trials on Gene Therapy and Vaccination

**DOI:** 10.3390/ijms25063151

**Published:** 2024-03-09

**Authors:** Ursa Lampreht Tratar, Tanja Jesenko, Masa Omerzel, Alenka Seliskar, Urban Stupan, Mihajlo Djokic, Jerneja Sredensek, Blaz Trotovsek, Gregor Sersa, Maja Cemazar

**Affiliations:** 1Department of Experimental Oncology, Institute of Oncology Ljubljana, 1000 Ljubljana, Slovenia; ulampreht@onko-i.si (U.L.T.);; 2Small Animal Clinic, Veterinary Faculty, University of Ljubljana, 1000 Ljubljana, Slovenia; alenka.seliskar@vf.uni-lj.si (A.S.);; 3Faculty of Medicine, University of Ljubljana, Vrazov trg 2, 1000 Ljubljana, Slovenia; 4Faculty of Pharmacy, University of Ljubljana, Askerčeva 7, 1000 Ljubljana, Slovenia; 5Department of Abdominal Surgery, University Medical Center Ljubljana, 1000 Ljubljana, Sloveniablaz.trotovsek@kclj.si (B.T.); 6Faculty of Health Sciences, University of Ljubljana, 1000 Ljubljana, Slovenia; 7Faculty of Health Sciences, University of Primorska, 6310 Izola, Slovenia

**Keywords:** interleukin 12, gene electrotransfer, porcine model, oncology, immunotherapy

## Abstract

Gene electrotransfer (GET) of plasmids encoding interleukin 12 (IL-12) has already been used for the treatment of various types of tumors in human oncology and as an adjuvant in DNA vaccines. In recent years, we have developed a plasmid encoding human IL-12 (phIL12) that is currently in a phase I clinical study. The aim was to confirm the results of a non-clinical study in mice on pharmacokinetic characteristics and safety in a porcine model that better resembled human skin. The GET of phIL12 in the skin was performed on nine pigs using different concentrations of plasmid phIL12 and invasive (needle) or noninvasive (plate) types of electrodes. The results of our study demonstrate that the GET of phIL-12 with needle electrodes induced the highest expression of IL-12 at the protein level on day 7 after the procedure. The plasmid was distributed to all tested organs; however, its amount decreased over time and was at a minimum 28 days after GET. Based on plasmid copy number and expression results, together with blood analysis, we showed that IL-12 GET is safe in a porcine animal model. Furthermore, we demonstrated that pigs are a valuable model for human gene therapy safety studies.

## 1. Introduction

Electroporation is a physical method using electric pulses that cause permeabilization of the cell [1]. When used in combination with cytostatic drugs such as bleomycin or cisplatin, the treatment is called electrochemotherapy. Electrochemotherapy has already been widely used in human and veterinary oncology in several types of cancer, achieving a good local antitumor response [2,3]. To increase the local effectiveness of the treatment and influence the systemic immune response, electrochemotherapy has been proposed to be combined with gene electrotransfer of plasmids encoding IL-12 [4]. IL-12 is a cytokine with an antitumor effect that acts via interferon gamma (IFNγ). It stimulates the action of cytotoxic T lymphocytes [5], which creates pores in the tumor cell membrane via perforin, consequently allowing granzyme B to enter the cells [6], where it causes programmed cell death of the tumor cell, apoptosis [7]. In addition, IL-12 reduces the growth of tumor vessels via IFNγ, which contributes to the antitumor effectiveness of IL-12 [8]. Studies have already been published demonstrating the antitumor efficacy of IL-12 in preclinical studies on mice [9,10,11], in clinical studies on spontaneous tumors in dogs [12], and in human studies on melanoma, where in addition to a local effect, they have also demonstrated a systemic effect on distant untreated tumors [13]. IL-12 GET has been predominantly used as an intratumoral or intradermal application [14,15]. The skin, as a target organ, has a great advantage over other tissues because the field of therapy is extensive and accessible. In addition, there are cells in different layers of the skin that can initiate a different response to transfection and expression of the therapeutic gene. Muscle cells located in the deeper layers of the skin and subcutaneous tissue have long-term expression because they divide slowly, while keratinocytes located in the upper layers have a shorter life span and therefore a short expression of the transgene [16,17]. Electrodes used for skin are mainly divided into two types: plate (noninvasive) or needle (invasive) [18]. The advantages for noninvasive electrodes are that the application of the electrodes itself is not painful, as it does not breach the skin, and the electrodes can be moved rapidly on the body [19]; however, it has been shown that penetrating electrodes induce higher levels of transgene expression [20,21,22]. Although combination therapy has proven to be effective, there is a need for the optimization of gene therapy to improve the treatment outcome, specifically optimization concerning the different concentrations of plasmid DNA, the use of invasive or noninvasive electrodes, and the use of plasmid DNA lacking the antibiotic resistance gene. The latter is implemented in Europe by the European Medicines Agency (EMA) due to safety concerns regarding the use of antibiotics during the production of plasmids [23,24]. In recent years, we have constructed several antibiotic-resistance-gene-free plasmids, including plasmids encoding mouse, canine, and human IL-12 orthologs [8,25]. A plasmid encoding human interleukin-12 (phIL12) is currently in a phase I clinical study enrolling patients with basal cell carcinoma of the head and neck (Clinicaltrials.gov: NCT05077033) [26]. In addition, IL12 can also be used as an adjuvant to different types of vaccines, including DNA vaccines [27,28]. Recently, our developed plasmid was evaluated in in vitro studies [29] and in a preclinical study of the mouse tumor model CT26, where a plasmid with a transcript for the mouse ortholog IL-12 was used due to its biological inactivity in mice, which demonstrated its biological activity, safety, and pharmacodynamic properties (data not published yet). The main aim of this study was to confirm the safety of phIL12 GET in a porcine animal model (because the homology between human and porcine IL-12 is high, over 85% [30], and the skin characteristics are similar between pigs and humans [31]) to support the transition of phIL12 GET into human clinical studies. Additionally, we also investigated the use of different phIL12 GET modalities (different plasmid DNA concentrations and the use of invasive or noninvasive electrodes) on IL12 expression in the skin. 

## 2. Results

### 2.1. IL-12 mRNA Expression and the Plasmid Copy Number Were Increased 7 Days after the Procedure, and the Use of Invasive Needles Resulted in Higher Expression of IL-12 at the Protein Level

IL-12 expression at the mRNA level was significantly higher 7 days after the procedure, compared to other time points, in all groups, except the 2 mg/mL group with plate electrodes. Interestingly, the differences between the type of electrodes or the different amounts of plasmid used were not significant. Furthermore, the highest amount of plasmid DNA was present in the skin 7 days after the procedure, and the difference was significant. However, we did not observe any significant difference between different procedure modalities (different plasmid doses or the use of invasive/noninvasive electrodes). The plasmid copy number at 14 days and 28 days was minimal and even nonexistent in some groups (14 days: 1 mg/mL plate and needle; 28 days: 1 mg/mL and 2 mg/mL plate) (Figure 1).

Furthermore, at the protein level, the highest amount of hIL-12 protein was observed in the 2 mg/mL plasmid group. Here, the use of needle electrodes resulted in a higher expression of the IL-12 protein than the use of plate electrodes. These differences were observed 14 days after the procedure and were statistically significant. However, differences in the IL-12 protein level were not observed at 7 days or 28 days after treatment or in the groups treated with 1 mg/mL plasmid (Figure 1).

### 2.2. Plasmid DNA Did Not Persist in the Majority of the Organs from Day 14 On

The results from the copy number analysis showed that plasmid DNA was distributed in all organs on day 7, except in subiliac lymph nodes. Depending on the concentration of plasmid used, the amount of plasmid DNA varied in different organs. The highest number of plasmid copies was observed 7 days after the procedure in the liver and lung of the 1 mg/mL group and in the spleen of the 2 mg/mL group. In all of these organs, the number of plasmids diminished over time and was minimal 28 days after the procedure. Additionally, the presence of plasmid DNA in the ovaries was not detected from day 14. Similar results to those observed in the ovary were observed in the heart, superficial dorsal cervical lymph node, and kidney. In the case of the eyes and brain, the plasmid DNA persisted at the same amount throughout the observation period, in the case of the brain in the 2 mg/mL group and in the case of the eye in the 1 mg/mL group (Figure 2).

### 2.3. Plasmid DNA Is Not Retained at the Injection Site

To determine whether plasmid DNA was retained at the injection site, skin swabs were collected from the phIL12 GET areas using a 2 mg/mL plasmid concentration. The plasmid was detected on day 0, just after phIL12 GET, and on day 7, but not on day 14 after GET. Namely, on day 14, the plasmid copy number was no longer significantly different from the values of the control negative skin swab. There was no significant difference between the electrodes, plate, or needle used at any of the analyzed time points (Figure 3). Skin swab analysis showed that the plasmid was not retained on the skin surface.

### 2.4. PhIL12 GET Did Not Have Any Additional Effect on Whole Blood or Serum Parameters over 28 Days

The results from the CBC analysis showed that no significant abnormal values were observed in the majority of the parameters analyzed. There were some differences in the platelet count; in all three groups, we observed a drop in the platelet count from days 1 to 3 after the procedure. Although the data from the 1 mg/mL and 2 mg/mL groups were out of range, a drop was also observed for the control group, which dropped from 609.33 to 472.33 (×109/L) on the second day. The platelet parameters from all three groups were normal again 7 days after the procedure (Appendix A). The percentage of eosinophils also increased, which was observed in the control group as well as in the plasmid group; therefore, eosinophils are not marked in Appendix A. The results of the biochemistry analysis of the coagulation factors showed no deviations in the treated pigs compared to the pigs from the control groups. Similar results were observed in the analysis of electrolytes, glucose, and BUN, where again no significant changes were observed in the control group and plasmid groups (Appendix A). The results from the analysis of creatinine showed that in the 1 mg/mL group, the values were elevated from 14 to 28 days (116.66 to 140.9 µmol/L). However, similar results were observed in the control group, where the values of creatinine also increased 21 and 28 days after the procedure, in contrast to the values before the treatment. In the case of inorganic phosphate, AP, total protein, and albumin, we did not observe any significant changes, and the values were in the control group range or were borderline. However, in the case of AST values, we observed a transient increase after the procedure, which again was also present in the control group (Appendix A). When observing ALT, we observed an increase in the values that breached the reference range from the control group; however, the increase in the plasmid groups was maximal at 20%, whereas in the control group, the increase was more than 20%. Overall, we did not observe any systemic toxicity following the GET of phIL12.

## 3. Discussion

Pigs have already been shown to be a good model for human metabolic, cardiovascular, and infectious diseases, xenotransplantation, neurological disorders, and oncology studies. The main purpose of this study was therefore to support and confirm the non-clinical phIL12 GET data for the ongoing phase I clinical trials. In particular, this study was designed to collect additional data on the safety of the IL-12 GET procedure as recommended for the clinical development of the new drug [26,29]. We decided on a porcine model, as pigs can serve as a valuable model for human IL-12 safety studies due to high homology between human and porcine IL-12 (over 85%) [30].

First, we tested the use of different electrodes. Our results showed that there was no difference observed between the use of invasive or noninvasive electrodes or the amount of plasmid DNA at the mRNA level; however, we did confirm that the expression of the transgene at the mRNA level was the highest 7 days after the procedure. Furthermore, we evaluated IL-12 expression at the protein level, and we observed the highest expression 14 days after treatment. Interestingly, the elevation of both parameters did not occur at the same time point, and it has already been proven that the mRNA levels do not correlate with protein levels after transfection of the transgene [32]. Additionally, we observed a higher expression of the IL-12 protein when using 2 mg/mL plasmid and invasive electrodes. Similar results were observed in a study on porcine skin using a reporter gene (GFP), where the researchers concluded that needle electrodes provide higher expression, as penetration of the stratum corneum led to a more homogenous field distribution at the DNA injection site [33].

The presence of plasmid in porcine skin was observed 7 days after the procedure and was statistically significantly higher compared to the later time points. The retention of plasmid DNA decreased significantly at 14 days, and at 28 days, plasmid DNA was present only in the plate (1 mg/mL and 2 mg/mL) groups. These results are similar to those gathered from veterinary studies, where the presence of plasmid DNA 7 days after the procedure was still evident but reduced over time and was almost nonexistent 28 days after the procedure [12]. Skin swab analysis from the phIL12 GET showed that the plasmid was not retained on the surface of the skin at the site of injection, irrespective of the electrodes used. This is one of the important safety issues, showing that plasmids do not pose environmental risks and are most likely quickly degraded by skin DNAses [34,35]. Furthermore, the plasmid was not retained at any of the time points in the samples of subiliac lymph nodes, which could mean that this lymph node was not a draining lymph node for the area where IL-12 GET was performed. The plasmid DNA also persisted in several organs predominantly in the 7 days after the procedure. The highest amount of plasmid DNA was found in the liver, lungs, and spleen, known as the “first pass” organs, 7 days after the procedure [36,37,38]. However, the amount of plasmid DNA in these organs was already reduced to a minimum of 14 days after the procedure. Additionally, the safety of the procedure was also evaluated by analyzing different blood cells and biochemical parameters. The results showed no significant deviation from the results obtained from the pigs in the control group. Furthermore, the results of the primary study showed no vascular complications in pigs and no pathologic findings in the histologic samples of the liver vessels after euthanasia of the pigs that could influence our IL-12 GET study [39]. 

The major limitation of this study is the number of animals used. To follow the 3R rule for the primary study, only nine animals could be enrolled in the GET studies. The presence of only one animal per group per time point made it impossible to evaluate the effects statistically. However, the aim of this study was to support the data from mice in a different animal species where the homology of IL-12 was high (over 85%). This was confirmed by this study. Moreover, in the scope of 3R, we performed other studies along with the current one that intervened with the blood results of our study, specifically the liver parameters, which we did consider when discussing and evaluating the blood results. Furthermore, when evaluating the local changes using different electrodes and plasmid amounts, we could perform technical repetitions; however, to evaluate the use of invasive or noninvasive electrodes for transgene transfection and to determine the right amount of plasmid used, additional studies should be performed.

In this study, we showed that phIL12 gene delivery to skin is feasible; the plasmid was successfully transferred into pig skin, where the IL-12 protein was produced. We also provided safety evidence on plasmid distribution, showing that there are minimal safety concerns regarding phIL12 gene therapy. phIL12 was not retained on the surface of the skin, where phIL12 GET was performed. Moreover, it did not persist in vital organs, especially the gonads, which additionally proves the safety aspect of a possible transmission of the transgene to the offspring.

IL-12 has already been widely used in clinical trials in different fields, mainly cancer therapy and vaccination. Currently, there are over 250 studies together, of which 77 are active and 204 studies are recruiting patients. IL-12 is used in the field of vaccination, mainly as an immune adjuvant for the HIV vaccine [40], and in cancer therapy for the treatment of different solid tumors, as well as hematological malignancies. Therefore, our results further support the use of IL-12 as a safe agent that could be used in oncology as an immune response stimulant or in the vaccination field as an immune adjuvant.

## 4. Materials and Methods

### 4.1. Animals

This study was conducted in accordance with ARRIVE 2.0 (Animal Research: Reporting of In Vivo Experiments) guidelines after approval by the National Ethics Committee and National Veterinary Administration (approval number U34401-2/2021/5); it was performed using 9 Landrace and Large White hybrid pigs (Sus scrofa domesticus), aged 10 weeks and weighing 32.72 ± 1.49 kg. The animals were vaccinated with a vaccine against Porcine circovirus type 2 (PCV2) and the bacterium Mycoplasma hyopneumoniae 4–6 weeks before they were purchased. In the scope of 3R, simultaneous studies were performed: the primary study “Pathohistological changes of v. portae anastomoses following electrochemotherapy”, “Pharmacokinetics of maropitant and levobupivacaine”, “Measurements of blood pressure using Doppler”, “Transversus abdominis plane block”, and “Expression of miRNA and proteins of heart cells stored in cool ischemia” [39]. The pigs used for this study were obtained from a licensed pig breeder 7 days before the start of the study. They were housed in pairs or threes in indoor straw bedded pens with an enriched environment. The pens also allowed visual and audible contact with other pigs. The pigs were exposed to a natural light–dark cycle, had unlimited access to tap water from nipple drinkers, and were fed commercial feed for grower pigs twice a day. They were monitored continuously by video surveillance and research staff for the first 24 hours after anesthesia; then, they were checked by staff 3 times daily. Animals were assessed for general vital signs, food intake, urine, fecal output, and need for analgesia.

Although this study is quite complex, we followed the 3R rule of the primary study and therefore only 9 animals were enrolled. Therefore, the systemic effects (plasmid copy number) were impossible to evaluate statistically as we only had one animal per group per time point.

### 4.2. Anesthesia

The anesthesia of the pigs was performed as previously described [39]. Briefly, pigs were premedicated with ketamine (Bioketan, Vetoquinol Biowet, Gorzow, Poland) 10 mg/kg, midazolam (Midazolam Torrex, Torrex Chiesi Pharma GmbH, Wien, Austria) 0.5 mg/kg, and medetomidine (Domitor, Orion Corp., Turku, Finland) 0.02 mg/kg intramuscularly and induced to anesthesia with isoflurane (Isoflurin; Vetpharma Animal Health S.L., Barcelona, Spain) administered through a face mask; after this, they were endotracheally intubated and anesthesia was maintained with isoflurane. Carprofen (Rycarfa, Krka, Slovenia) 4 mg/kg was administered intravenously, and a fentanyl transdermal patch (Durogesic Janssen Pharmaceutica, Beerse, Belgium) of 75 μg/h was applied on the lateral aspect of the thorax. Carprofen was administered orally once daily for the next 6 days.

### 4.3. Blood Sampling: Hematology and Biochemistry

For blood sampling, a 16-gauge permanent central venous catheter was surgically placed into the external jugular vein which led subcutaneously to the back of the neck. Blood was drawn before the procedure and on days 1, 2, 3, 7, 14, 21, and 28. At each sampling, a complete blood count (CBC) with white blood cell differential count was performed using a hematology analyzer (Advia 120, Siemens, Munich, Germany). Additionally, biochemistry parameters together with coagulation parameters were measured using an automated biochemistry analyzer (RX Daytona+, Randox, Crumlin, United Kingdom). The blood and serum range parameters were set according to the blood and serum ranges of the control group, due to the fact that the primary study involved manipulation with the liver and liver vasculature and some changes in the liver biochemistry parameters of liver function were observed in the control groups (where only the primary study was performed and not IL-12 GET).

### 4.4. Study Design

The animals were divided into three groups with different plasmid DNA concentrations (phIL12) [28]: 0 mg/mL (0.9% NaCl), 1 mg/mL, and 2 mg/mL. The IL-12 GET was performed one time only on day 0 on 6 different areas measuring 1 cm in diameter (A–F) on the right flank of the animal (Figure 4). The upper three spots (A–C) were treated using noninvasive plate electrodes (with the distance between plates being 8 mm) and the lower spots (D–F) were treated with invasive needle electrodes (needle rows of a 4 mm distance) (Figure 4). Briefly, 0.5 mL of the plasmid DNA solution or NaCl was injected intradermally, and immediately afterwards, electric pulses (8 pulses, 1300 V/cm, 100 μs, 5 kHz) were applied using a generator of electric pulses (Cliniporator^TM^, IGEA s.r.l., Carpi, Italy). In total, the pigs receiving 1 mg/mL plasmid DNA on one spot received 3 mg of plasmid and the pigs receiving 2 mg/mL plasmid DNA per spot received a total of 6 mg of plasmid DNA. After GET, sutures were made in the centers of the plasmid injection to mark the treated areas for longer time intervals (Figure 4).

### 4.5. Euthanasia and Sampling

Animals were euthanized at three different time points: 7, 14, and 28 days after procedures, with one animal per group (Figure 5). Pigs were first anesthetized in the same way as at the beginning of this study, and then euthanasia was performed by intravenous injection of T61 euthanasia solution (Intervet, Boxmeer, The Netherlands). After euthanasia, the skin flap where IL-12 GET was performed was excised. Then, each spot (1 cm in diameter) was separately excised, divided into three parts, and placed into 1.5 mL tubes at −80 °C for further downstream analysis of IL-12 expression. Furthermore, 4 mm punch biopsies were taken from each of the following organs, eye, brain, lungs, heart, liver, spleen, kidney, ovary, and draining lymph nodes (lnn. Subiliaci, lnn. Cervicales superficiales dorsales), and snap-frozen. Skin swabs were collected and analyzed only in pigs receiving 2 mg/mL phIL12 on the day of IL-12 GET and 7 and 14 days after GET. Swabs were collected from the IL-12 GET areas using eNAT^®^ nucleic acid collection and preservation tubes (Copan Diagnostics, Murrieta, CA, USA). On the same day, the samples were vortexed, transferred to Eppendorf tubes, and stored for further analysis at −80 °C.

### 4.6. Isolation of RNA and DNA

The frozen skin and organ samples were homogenized using a mortar and pestle and weighed to obtain 25–30 mg of the tissue for total RNA extraction and 10–15 mg for total DNA extraction. Total RNA was extracted using the TRIzol Plus RNA Purification System (Life Technologies, Carlsbad, CA, USA) and peqGOLD Total RNA kit (VWR International, Radnor, PA, USA) according to the manufacturer’s instructions. Thereafter, 750 ng of isolated RNA was reverse-transcribed (Primus 25 advanced^®^ Thermocycler, VWR) into cDNA using the SuperScript VILO cDNA Synthesis Kit (Life Technologies). Total DNA from tissue was isolated using a DNeasy Blood & Tissue kit (Qiagen, Hilden, Germany) as described previously [30] and for skin swabs it was isolated using a Genelute™ Plasma/Serum Cell-Free Circulating DNA Purification Midi Kit.

### 4.7. qRT-PCR

SYBR Green chemistry (PowerUP^TM^ SYBR^TM^ Green Master Mix (2×), Thermo Fisher Scientific, Waltham, MA, USA) was used for qRT-PCR. Each qRT-PCR reaction consisted of a 20 µL reaction mixture containing 10 ng of cDNA (transgene expression) or 10 ng of total DNA (plasmid copy number). For the determination of gene expression, the cycling conditions were as follows: 2 min at 50 °C, 2 min at 95 °C, 40 cycles of 15 s at 95 °C, and 1 min at 60 °C; for melting curve determination, the conditions were 15 s at 95 °C, 1 min at 60 °C, and 15 s at 95 °C. Furthermore, for the determination of plasmid copy number, the cycling conditions were as follows: 2 min at 50 °C, 2 min at 95 °C, 40 cycles of 15 s at 95 °C, 30 s at 58 °C, and 30 s at 72 °C; for melting curve determination, the conditions were 15 s at 95 °C, 1 min at 60 °C, and 15 s at 95 °C. The level of transgene expression (mRNA) in skin samples after GET was determined by relative quantification (QuantStudio 3, Thermo Fisher Scientific) using reference (housekeeping) genes for porcine BA and B2M (Table 1). Primer oligonucleotides for IL-12 were prepared so that they were specific to our transgene, meaning that they only targeted the IL-12 produced by cells after GET. The specific primers did not allow for amplification of endogenous human or porcine IL-12 mRNA or any other known dsDNA sequence [29]. Absolute quantification (QuantStudio 3, Thermo Fisher Scientific) was used to determine the number of plasmid DNA copies in tissue samples after GET and skin swabs, as described previously [29]. Briefly, the method was based on a standard curve from a known number of copies of synthetic double-stranded DNA (gBlocks, Integrated DNA Technologies, Coralville, IA, USA) and serial dilutions. The plasmid copy number was then normalized to 1 µg of the total DNA sample from tissues or 0.1 ng of plasmid DNA extracted from skin swabs. Primer oligonucleotides specific to plasmid p21-hIL-12-ORT (phIl12) were prepared as described previously [29]. As a negative control for qRT-PCR, a no-template control (NTC), including all the PCR reagents except for the template (cDNA), was used. All the samples were run in technical duplicates.

### 4.8. ELISA hIL-12

IL-12 protein from skin tissue was evaluated using ELISA. First, the proteins were isolated from the frozen samples using an RIPA lysis buffer system (sc-24948, Santa Cruz Biotechnology, Dallas, TX, USA) according to the manufacturer’s instructions. Briefly, 1 mL of 1× RIPA buffer containing 10 μL PMSF solution, 10 μL sodium orthovanadate solution, and 10–20 μL protease inhibitor cocktail was used to prepare the complete RIPA solution. Then, 3 mL of complete RIPA buffer was used per 1 g of tissue and mixed thoroughly. The mixture was incubated for 15–30 min on ice and centrifuged at 12,000× *g* for 15 min at 4 °C. The supernatant was removed and further analyzed by ELISA (Human IL-12 p70 Quantikine ELISA Kit R&D System, Minneapolis, MN, USA) to detect human IL-12.

### 4.9. Statistical Analysis

The results from the skin samples where local changes after IL-12 GET were observed were analyzed using GraphPad Prism 9 (GraphPad software, San Diego, CA, USA) software. The data were checked for normality using the Shapiro–Wilk test. Significance was determined by two-tailed t test or one-way ANOVA in the case of skin samples. A *p* value less than 0.05 was considered significant. The results from organs and skin swabs could not be statistically analyzed, as there were no biological replications due to the small number of experimental pigs (3R).

## Figures and Tables

**Figure 1 ijms-25-03151-f001:**
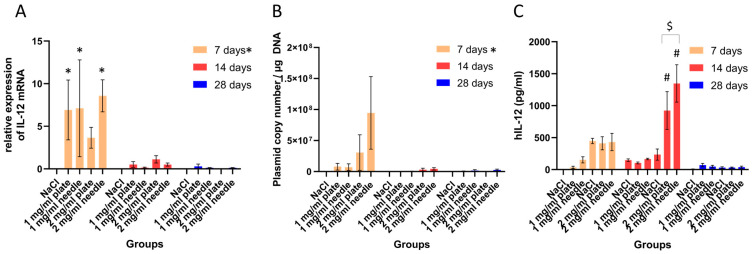
The level of relative mRNA expression (**A**), plasmid copy number (**B**), and protein level (**C**) in the skin with different procedure modalities at different time points. * represents *p* < 0.05 when analyzing the groups by day and analyzing the same groups on days 14 and 28 (for relative expression of IL-12 mRNA). # indicates *p* < 0.05 when compared to other groups on day 14 and the same groups on days 7 and 28. $ represents *p* < 0.05 when comparing the 2 mg/mL plate and needle groups at 14 days.

**Figure 2 ijms-25-03151-f002:**
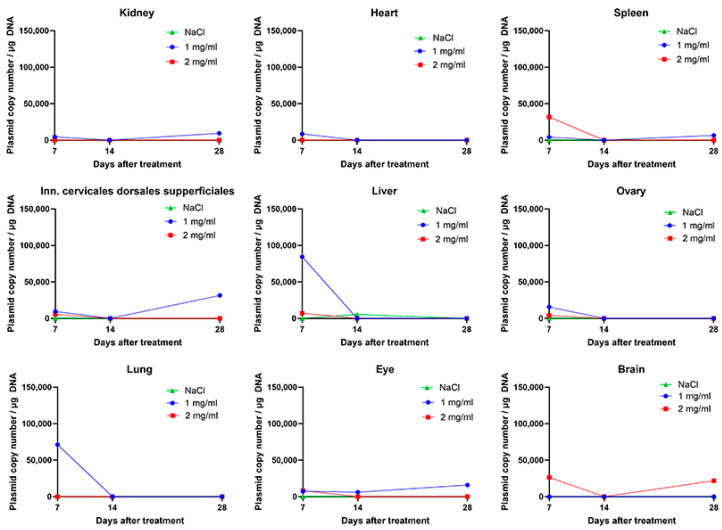
Copy number analysis of different organs after 1 mg/mL or 2 mg/mL phIL12 GET at three different time points.

**Figure 3 ijms-25-03151-f003:**
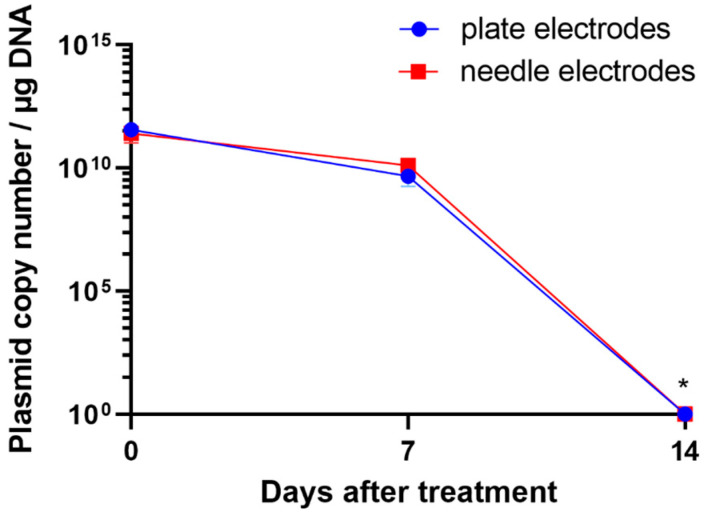
Plasmid copy number in skin swabs from the phIL12 GET area. * *p* < 0.05 significantly different than on days 0 and 7.

**Figure 4 ijms-25-03151-f004:**
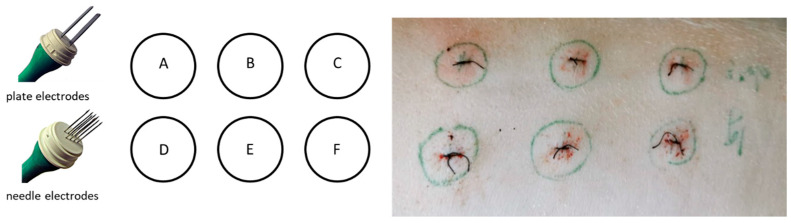
The IL-12 GET was performed on 6 spots per animal. (**A**–**C**) are with noninvasive plate electrode and (**D**–**F**) are with invasive needle electrode.

**Figure 5 ijms-25-03151-f005:**
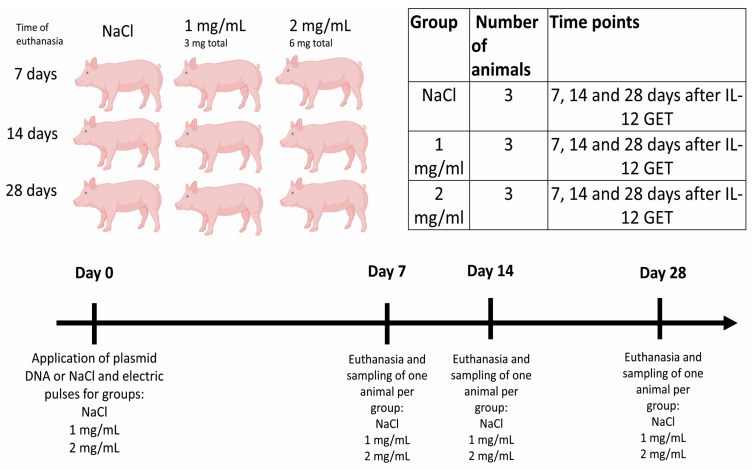
Schematic presentation of the animals in different groups and timeline of the procedure and sampling. Created using BioRender.com (Accessed on 7 September 2023).

**Table 1 ijms-25-03151-t001:** List of the primers used for RT-qPCR.

Primer	Primer Details	Sequence
Transgene expression		
pBA forward	Porcine internal/housekeeping expression control (beta actin)	TCCACGAAACTACCTTCAACTC
pBA reverse	Porcine internal/housekeeping expression control (beta actin)	GATCTCCTTCTGCATCCTGTC
pB2M forward	Porcine internal/housekeeping expression control (beta 2-microglobulin)	CCACACTGAGTTCACTCCTAAC
pB2M reverse	Porcine internal/housekeeping expression control (beta 2-microglobulin)	GGTCTCGATCCCACTTAACTATC
hIL-12 forward	Expression primer: specific to the linker region between the p40 and p35 IL-12 subunits (phIL12)	CTGCAGTGTTCCTGGAGTAG
hIL-12 reverse	Expression primer: specific to the linker region between the p40 and p35 IL-12 subunits (phIL12)	GAACATTCCTGGGTCTGGAG
Copy number		
phIL12 forward	Copy number primer: specific to the plasmid backbone (ori)	GCAGAGCGCAGATACCAAATA
phIL12 reverse	Copy number primer: specific to the plasmid backbone (ori)	GCGCCTTATCCGGTAACTATC

## Data Availability

The data presented in this study are available upon request from the corresponding author.

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
