# Peer review of "Safety and Efficacy of IL-12 Plasmid DNA Transfection into Pig Skin: Supportive Data for Human Clinical Trials on Gene Therapy and Vaccination"

_ijms, 2024, doi:10.3390/ijms25063151_

Round 1
Reviewer 1 Report
Comments and Suggestions for Authors
In this study, the authors showed transfection of plasmid including IL-12 into pigs and they showed that the transcription and tranlational level of IL-12 were maintained in the first 7 days and then reduced after this.
Unforuntalry, the design and interpretation of most experiments is not accurate leading to difficulty of reaching a solid conclusion,
Major concerns
1- Figure 1 is completely not convincing: regarding the methology
a) What is the initial number of PBMCs included? counting the cells are not the accurate measurement of proliferation? IL-2 is T-cell polyclonal, but the author did not characterize the percentage of T-cell in the PBMCs. Since it is a positive control, it is not clear that IL-2 and Il-12 showed the same efficacy. More surprisngly, these PBMCs isolated from pigs, so if IL-12 induce cell proliferation due to binding with porcine Il-12 receptor, how can the author explain that human IL-12 showed the same efficacy of porcine IL-12 on the porcine PBMCs?
2) Figure 2: I could not trust the data of this figure due to the following: the primers mentioned by the authors are not accurate.
a) sequences of porcine B-actin and B2M mentioned by the authors are the same. They target porcine B2M , not B actin
b) Human IL-12 is not accurate, the same for poricne IL-12
3- Data could not be matched. For example; it is not clear why some doses of IL-12 (2mg/ul) after liver function profiles, while no plasmid DNA was detected in the liver.
Comments on the Quality of English LanguageExtensive language editing is required
Author Response
- What is the initial number of PBMCs included? counting the cells are not the accurate measurement of proliferation? IL-2 is T-cell polyclonal, but the author did not characterize the percentage of T-cell in the PBMCs. Since it is a positive control, it is not clear that IL-2 and Il-12 showed the same efficacy. More surprisngly, these PBMCs isolated from pigs, so if IL-12 induce cell proliferation due to binding with porcine Il-12 receptor, how can the author explain that human IL-12 showed the same efficacy of porcine IL-12 on the porcine PBMCs?
Thank you for your comment.
The initial number of PBMC before stimulation was 1 x 10Ë„6 per well as stated in Materials and Methods section. We were specifically interested in the increasing number of cells after the addition of the proliferation agent, and therefore counting the exact number of cells was the method of choice. PBMC counting is a traditional way of PBMC proliferation assessment as the number of cells is indeed the direct measure of proliferation. Several other simple methods are available; however, they frequently measure more indirect evidence of cell proliferation such as metabolic activity (MTS, PrestoBlue etc.). In line with the reviewer comment, we agree that more detailed methods for determination of proliferation of different types of blood cells are available such as flow cytometry or evaluation of DNA synthesis using BrdU (5-Bromo-2´-Deoxyuridine) or EdU (5-ethynyl-2’-deoxyuridine). However due to limited resources we focused on a more basic method to confirm the biological activity of human IL-12 compared to pig IL-12. IL-2 is indeed a proliferating agent for T cells, but so is IL-12, which was the agent in question. Furthermore, the main question of this part of the study was whether human IL-12 has the same level of stimulation as porcine IL-12. This was indeed confirmed.
We added the sentences in a Material and Methods (Biological activity of human IL-12 in pig PBMCs): The addition of porcine IL-12 (912-PL, Recombinant Porcine IL-12 Protein, 1 mg/ml, R&D System, Minneapolis, Minnesota, USA) and IL-2 (1 mg/ml, Miltenyi Biotec, Bergisch Gladbachu, Germany) to cells was used as a positive control (IL-12 was used as overall positive control as it is a potent proliferating agent and porcine IL-12 as a positive control for human IL-12). The negative control presents cells without the addition of interleukins. On day 2 and day 4 after stimulation with interleukins, the number of cells was counted using a Corning CytoSmart Cell Counter (Corning, Somerville, Massachusetts, USA) to determine PBMC proliferation as an indicator of biological activity as PBMC counting is a traditional way of PBMC proliferation assessment and the number of cells is indeed the direct measure of proliferation.
2) Figure 2: I could not trust the data of this figure due to the following: the primers mentioned by the authors are not accurate.
- sequences of porcine B-actin and B2M mentioned by the authors are the same. They target porcine B2M , not B actin
Thank you for your comment, it was a typo mistake and the sequence for BA is now corrected:
pBA forward Porcine internal/housekeeping expression control (beta actin)
TCCACGAAACTACCTTCAACTC
pBA reverse Porcine internal/housekeeping expression control (beta actin)
GATCTCCTTCTGCATCCTGTC
- b) Human IL-12 is not accurate, the same for poricne IL-12
Thank you for your comment. The primers for the copy number phIL12 (reverse and forward) are not IL12 specific, but plasmid backbone specific, targeting the ori site.
Similar is for the primers for the transgene, as it was cruical that the primers targets only the transgene of our plasmid and not the endogenous IL-12. Our transgene IL-12 is distinct from the native IL-12 in a way that p40 and p35 IL-12 subunits are linked through a nucleotide linker region, so also a mRNA for our transgene is comprised of a single mRNA. The primers for IL-12 are designed to target the linker, which is a unique feature detecting only our transgene IL-12. Below is the map and sequences of the primers (termed “over-linker”).
These sentences were added to the text:
»Primer oligonucleotides for IL-12 were prepared so that they are specific for our transgene, meaning they target only the IL-12 produced by cells after the GET. The specific primers do not allow amplification of endogenous, either human or porcine IL-12 mRNA or any other known dsDNA sequence [29]«.
3- Data could not be matched. For example; it is not clear why some doses of IL-12 (2mg/ul) after liver function profiles, while no plasmid DNA was detected in the liver.
Thank you for your comment. The primary study involved manipulation with the liver and liver vasculature, therefore we monitored all the important biochemical parameters of liver function as well as kidney function, glucose, etc… As the changes observed in the liver biochemistry were present in control groups (where only primary study was done and not IL-12 GET) as well as in our therapeutic groups (1 mg/ml and 2 mg/ml IL-12 GET groups) we contributed these changes to the primary study and not to our experiments.
The sentence clarifying the situation was added to the Material and methods:
The blood and serum range parameters were set according to the blood and serum ranges of the control group, due to the fact that the primary study involved manipulation with the liver and liver vasculature and some changes of liver biochemistry parameters of liver function was observed in control groups (where only primary study was performed and not IL-12 GET).
And in the discussion part it is already stated as one of the limitations of the study:
Moreover, in the scope of 3R, we performed other studies along with ours that intervened with the blood results of our study – specifically the liver parameters, which we did con-sidered when discussing and evaluating the blood results.
Comments on the Quality of English Language
Extensive language editing is required.
The paper was checked again for the English grammar by the AJE grammar check tool which is acknowledged in the Acknowledgment section.

Reviewer 2 Report
Comments and Suggestions for Authors
The objective of the study was to confirm the safety of phIL12 GET in a porcine animal model in which it is biologically active to support the transition of phIL12 GET into human clinical studies. Additionally, the authors studied the use of different phIL12 GET modalities (different plasmid DNA concentrations and use of invasive or noninvasive electrodes) on IL12 expression in the skin.
Major issues.
-The authors should include in the justification of the study the perceived clinical benefits from the potential findings of this work.
-Please include a table with summary of the experimental design (groups and treatment per group) to make clear the differences and also please include a graph with timeline of the animal phase of the study to present the procedures temporally.
-To note that 9 animals is an extremely small number of animals for such a complex experiment and this should be mentioned repeatedly in the study to make this very clear to readers.
-For the PCRs, please mention all the conditions of the assays, not just the primers.
-Statistics: why using parametric techniques? There is no mention anywhere about normal distribution of data…..
Minor issues.
-Tables 1-3 are too large and can be moved to supplementary material.
-The Discussion does not fully cover all the work performed and does not explain all the findings, so it should be extended.
-Can you please provide the vaccination schedule of these pigs, please? I shall raise a concern in the next round of assessment, based on the vaccines performed to the animals.
Author Response
Please include a table with summary of the experimental design (groups and treatment per group) to make clear the differences and also please include a graph with timeline of the animal phase of the study to present the procedures temporally.
Thank you for your comment. We added additional table and timeline in figure 5.
-To note that 9 animals is an extremely small number of animals for such a complex experiment and this should be mentioned repeatedly in the study to make this very clear to readers.
Thank you for your comment. Next, to mentioning this in Discussion (The major limitation of the study is the number of animals used. To follow the 3R rule for the primary study, only 9 animals could be enrolled in the GET studies. The presence of only one animal per group per time point makes it impossible to evaluate the effects statistically) and in statistical analysis in Material and methods (The results from organs and skin swabs could not be statistically analyzed, as there were no biological replications due to the small number of experimental pigs (3R), we also added: “Although this study is quite complex, we followed the 3R rule of the primary study and therefore only 9 animals were enrolled. Therefore, the systemic effects (plasmid copy number) were impossible to evaluate statistically as we only had one animal per group per time point.” in Material and methods section (Animals)
-For the PCRs, please mention all the conditions of the assays, not just the primers.
Thank you for your comment. We added all conditions for the PCR in the Material and Methods:
Each qRT-PCR reaction was a 20 µL reaction mixture containing 10 ng of cDNA (transgene expression) or 10 ng of total DNA ( plasmid copy number). For the determination of gene expression, the cycling conditions were: 2 min at 50 â—¦C, 2 min at 95 â—¦C, 40 cycles of 15 s at 95 â—¦C, 1 min at 60 â—¦C; for melting curve determination, 15 s at 95 â—¦C, 1 min at 60 â—¦C, 15 s at 95 â—¦C. Furthermore, for the determination of plasmid copy number, the cycling conditions were: 2 min at 50 â—¦C, 2 min at 95 â—¦C, 40 cycles of 15 s at 95 â—¦C, 30 s at 58 â—¦C and 30 s at 72 â—¦C; for melting curve determination, 15 s at 95 â—¦C, 1 min at 60 â—¦C, 15 s at 95 â—¦C-
Statistics: why using parametric techniques? There is no mention anywhere about normal distribution of data…..
Thank you for your comment. The data was checked for normal distribution and the sentence was added: The data were checked for normality by Shapiro-Wilk test.
Minor issues.
-Tables 1-3 are too large and can be moved to supplementary material.
Thank you for your comment. We moved Tables 1-3 to Supplementary material
-The Discussion does not fully cover all the work performed and does not explain all the findings, so it should be extended.
Thank you for your comment. We added two paragraphs to the discussion:
The main purpose of this study was therefore to support and confirm the non-clinical phIL12 GET data for the ongoing Phase I clinical trials. In particular, this study was designed to collect additional data on the safety of the IL-12 GET procedure as recommended for the clinical development of the new drug in animal species in which hIL-12 is biologically active. [26], [29]. For this reason, we first confirmed that porcine PBMCs are equally stimulated by human IL-12 and porcine IL-12, showing that pigs could serve as a valuable model for human IL-12 safety studies.
Additionally, the safety of the procedure was also evaluated by analyzing different blood cells and biochemical parameters. The results showed no significant deviation from the results obtained from the pigs in the control group. Furthermore, the results of the primary study showed no vascular complications in pigs and no pathologic findings in the histologic samples of the liver vessels after euthanasia of the pigs that could influence our IL-12 GET study. [36]
-Can you please provide the vaccination schedule of these pigs, please? I shall raise a concern in the next round of assessment, based on the vaccines performed to the animals.
Thank you for your comment. We added further clarification on a time of the GET application in Material and methods – Study design: “The IL-12 GET was performed one-time only on day 0 on 6 different areas measuring 1 cm in diameter (A-F) on the right flank of the animal (Figure 4).” In addition, a new Figure 5 is now included in the manuscript, where exact time schedule of our experiments is presented.
Additionally, we want to point out that this experiment was not a vaccination of the pigs, but a one-time application of plasmid IL-12 with electric pulses (GET) in order to cause local increase of the transgene and consequently boost the immune response by activating theTh1 response. Therefore, we did not use any vaccination schedules as this was one time application.
Round 2
Reviewer 1 Report
Comments and Suggestions for Authors
The authors provided answers to my concerns, however, these answers were not convincing, especially counting the PBMCs.
Comments on the Quality of English LanguageModerate language editing
Author Response
Thank you for your comment. We have additionally added references in Material and methods, where it states that the concentration of isolated PBMCs is traditionally measured by manual counting with a hemacytometer:
Weinberg A., Zhang L., Brown D., Erice A., Polsky B., Hirsch M.S., Owens S., Lamb K. Viability and Functional Activity of Cryopreserved Mononuclear Cells. Clin Diagn Lab Immunol 2000, 7(4), 714–716.doi: 10.1128/cdli.7.4.714-716.2000
Li-Ying Chan L., Laverty D.J., Smith T., Nejad P., Hei H., Gandhi R., Kuksin D., Qiu J. Accurate measurement of peripheral blood mononuclear cell concentration using image cytometry to eliminate RBC-induced counting error. J Immunol Methods 2013, 388, 25-32. doi: 10.1016/j.jim.2012.11.010.
Reviewer 2 Report
Comments and Suggestions for Authors
-Can you please provide the vaccination schedule of these pigs, please? I shall raise a concern in the next round of assessment, based on the vaccines performed to the animals.
Please provide the vaccination schedule of the experimental animals with commercial vaccines (e.g., PRRSV, atrophic rhinitis, Mycoplasma, etc.)
This was not answered and this raises significant suspicions.
Author Response
Thank you very much for your comment. We are sorry that we did not respond sooner, there was a misunderstanding with the question. The animals were vaccinated with the vaccine against: Porcine circovirus type 2 (PCV2) and the bacterium Mycoplasma hyopneumoniae 4–6 weeks before they were purchased. The sentence was added in the Material and methods section: Animals.
Round 3
Reviewer 1 Report
Comments and Suggestions for Authors
The authors provided references, however they are not matched with my concerns or authors' design.
Simply: The authors claimed that PBMCs proliferation using IL-12, I have concerned about proliferation of PBMCS and how the authors confirmed the proliferation. PBMCs is a collection of different cells; monocytes lymphocytes, neutrophils, NK cells. Monocytes, neutrophils and NK cells are not proliferating and only lymphocytes can proliferate by some agents or microbes such as cytomegalovirus.
The first reference showed that lymphocytes can be proliferated by CMV , while the second reference about counting PBMCs without RBCs contaminations.
IL-12 can affect the activity of PBMCs by affecting cytokine release, but I could not find references for using IL-12 as a proliferating for PBMCs
Comments on the Quality of English LanguageModerate language editing
Author Response
Dear reviewer, thank you for your comment.
We understand your concerns regarding the use of IL-12 as a proliferating agent for the PBMC and with the choice of methods that we used. We agree that more detailed methods for determination of proliferation of different types of blood cells are available such as flow cytometry or evaluation of DNA synthesis using BrdU (5-Bromo-2´-Deoxyuridine) or EdU (5-ethynyl-2’-deoxyuridine). However due to limited resources we focused on a more basic method.
For this reason, we decided to remove the first part of the study and to cite some additional references that already showed that human and porcine IL-12 are highly homologous.
Thank you

Reviewer 2 Report
Comments and Suggestions for Authors
No further comments.
Author Response
Thank you.
Round 4
Reviewer 1 Report
Comments and Suggestions for Authors
I am not confident about the story now. Since I saw the first version and last version, I felt the deletion of many parts of the manuscript makes the home message is not solid. May be other reviewers could accept it.
Comments on the Quality of English LanguageModerate language editing